# High-Grade Infection after Branched Endovascular Aortic Repair in Patient with Recent COVID-19 Hospitalization

**DOI:** 10.3390/diagnostics14020205

**Published:** 2024-01-18

**Authors:** Alireza Mohseni, Alessia Di Girolamo, Rocco Cangiano, Marta Ascione, Luca di Marzo, Wassim Mansour

**Affiliations:** Department of General Surgery and Surgical Specialties, “Sapienza” University of Rome, Policlinico Umberto I, Viale del Policlinico, 155, 00161 Rome, Italy; mohseni.1974637@studenti.uniroma1.it (A.M.); alessia.digirolamo@uniroma1.it (A.D.G.); rocco.cangiano@uniroma1.it (R.C.); marta.ascione@uniroma1.it (M.A.); luca.dimarzo@uniroma1.it (L.d.M.)

**Keywords:** COVID-19, thoracoabdominal aortic aneurysm, post-intervention complications, bacterial superinfection

## Abstract

In the context of the COVID-19 pandemic, the global healthcare landscape has undergone significant transformations, particularly impacting the management of complex medical conditions such as aortic aneurysms. This study focuses on a 76-year-old female patient with a history of extensive cardiovascular surgeries, including aortic valve replacement, Bentall operation, and Frozen Elephant Trunk procedure, who presented with a type II thoracoabdominal aortic aneurysm post-COVID-19 recovery. A comprehensive frailty assessment using the Modified Frailty Index and a two-phase endovascular approach for aneurysm treatment, considering the patient’s frailty and complex medical history was performed. Upon successful aneurysm management, the patient’s postoperative course was complicated by COVID-19 reinfection and Enterococcus faecalis superinfection, highlighting the increased risk of bacterial superinfections and the challenges posed by antimicrobial resistance in COVID-19 patients. The study underscores the necessity of vigilant postoperative surveillance and a multidisciplinary approach in managing such complex cases, highlighting the importance of personalized care strategies, integrating cardiovascular and infectious disease management, and adapting healthcare practices to the unique challenges of the pandemic. This case contributes to the evolution of knowledge on managing aortic aneurysms in the COVID-19 era, advocating for patient-centric treatment approaches and continuous research into long-term patient outcomes.

**Figure 1 diagnostics-14-00205-f001:**
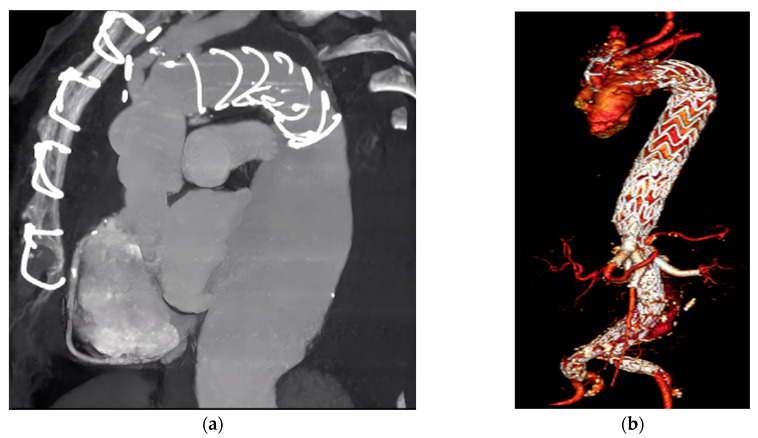
Complex aortic aneurysm management in a senior patient post-COVID-19: integrating surgical history with contemporary endovascular techniques. Panel (**a**) presents the angiography-CT scan outcomes of a 76-year-old female patient, admitted in February 2022 following her discharge from a COVID-19 ward. The patient’s extensive medical history includes systemic arterial hypertension, paroxysmal atrial fibrillation, and chronic ischemic cerebrovascular disease. Diagnostic imaging revealed a type II thoracoabdominal aortic aneurysm, characterized by significant dilation in the descending thoracic aorta, with a maximum diameter of 72 mm. This patient’s surgical history is particularly notable, encompassing aortic valve and ascending aorta replacement, a Bentall operation, and a Frozen Elephant Trunk (FET) procedure. The complexity of her medical history, including a recent sternotomy and aortic valve replacement, is evident in the postoperative findings. The patient’s case exemplifies the intricate challenges in aortic aneurysm management during the COVID-19 pandemic, marked by the emergence of bacterial co-infections and super-infections, and the systemic inflammatory response induced by COVID-19 depicted in a later figure [1,2]. Panel (**b**) illustrates the later-deployed endoprostheses, employed to exclude the type II Thoracoabdominal Aortic Aneurysm. The pandemic era’s widespread use of antibiotics, including aminoglycosides, β-lactams, and β-lactamase inhibitors, has led to an escalation in antimicrobial resistance rates, posing significant concerns in aortic aneurysm management [3]. Thus, given the patient’s advanced age and intricate medical history, a frailty assessment was imperative, particularly considering her recent transfer from a COVID-19 ward [4]. This assessment influenced the choice of a less invasive endovascular strategy over open surgery [5]. The initial phase involved a combination of endografts, including the COOK thoracic endograft, a T-Branch branched endograft, and a COOK Unibody endograft (all from Cook Inc., Bloomington, IN, USA) conducted under general anesthesia and percutaneous access. One month later, during the subsequent phase, the intervention was completed by deploying covered stents to bridge the branched endograft and the target vessels, employing VBX (WL Gore & Associates, Flagstaff, AZ, USA) endoprostheses. This strategic approach, culminating in the comprehensive exclusion of the aneurysm, underscores the evolving strategies in managing complex aortic aneurysm cases in the COVID-19 era, where cardiovascular and infectious disease considerations intersect [6,7]. The figure synthesizes the multifaceted challenges and strategic adaptations in aortic aneurysm management during these unprecedented times, highlighting the need for comprehensive and integrated patient care.

**Figure 2 diagnostics-14-00205-f002:**
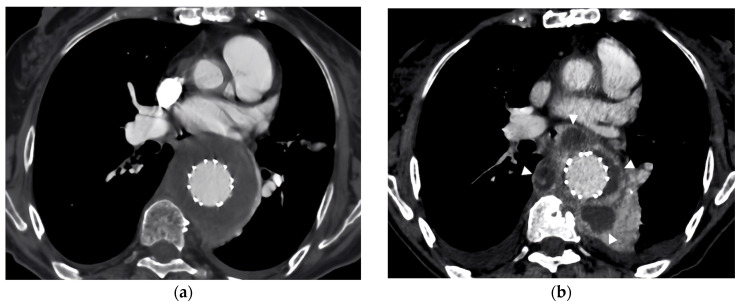
Evolving challenges in postoperative care: CT-angiography insights into aortic aneurysm management and COVID-19 complications. Panel (**a**) illustrates the successful postoperative exclusion of the aneurysmatic sac following the deployment of covered stents to bridge between the branched endograft and the target vessels. Subsequent angiography confirmed the complete exclusion of the aneurysm and demonstrated excellent patency of visceral arteries [8,9]. This phase of the intervention was crucial in ensuring the effectiveness of the endovascular strategy, as evidenced by a follow-up CTA scan conducted four days post-intervention, which validated the successful exclusion of the aneurysm with maintained patency. Despite prior prophylactic measures, the patient had experienced a post-operative reinfection with COVID-19, necessitating an extended period of intubation and an 8-day stay in the ICU. Apart from this, the patient’s recovery was uneventful. Following her successful vascular intervention, 6-months post operation, she was hospitalized again due to high-grade fever, non-specific pain, and asthenia, raising concerns about a potential underlying condition. A further CTA revealed the appearance of septic fluid collection sacs as depicted in Panel (**b**) with pointers. A thorough evaluation revealed symptoms consistent with bacterial sepsis, and comprehensive diagnostic procedures, including a direct fluid sample culture, confirmed an Enterococcus faecalis infection [1]. This infection was managed through a targeted antibiotic regimen, highlighting the patient’s susceptibility to bacterial superinfections, potentially exacerbated by her recent COVID-19 infection [10]. The management of this case aligns with existing research that emphasizes the complexities of managing cardiovascular conditions during the COVID-19 pandemic [2,6,11]. The systemic effects of COVID-19, notably on coagulation pathways and inflammatory responses, present additional risks for patients with aortic aneurysms [2]. This situation necessitates an integrated care approach that adeptly addresses both COVID-19 and cardiovascular health. The pandemic has significantly impacted healthcare practices, particularly in the management of aortic aneurysms, demonstrating the necessity for adaptable care strategies for patients with pre-existing cardiovascular conditions [1,2]. The rise in bacterial co-infections among COVID-19 patients adds complexity, requiring proactive and targeted infection control measures [3]. Furthermore, it has highlighted the significance of a multidisciplinary and patient-centric approach in treating frail patients with multiple comorbidities. The choice of a less invasive endovascular approach in this patient reflects a shift towards personalized treatment strategies [12,13]. This approach is crucial for minimizing surgical risks and optimizing patient outcomes, particularly during a period marked by evolving healthcare resources and priorities. The case serves to underscore the evolving nature of cardiovascular care, advocating for the prioritization of patient-centric strategies and adaptability to the changing healthcare environment. This figure encapsulates the multifaceted challenges in postoperative care and monitoring, emphasizing the importance of vigilant follow-up in patients with aortic aneurysms, especially those with a history of COVID-19 and its potential complications.

## Data Availability

No new data were created or analyzed in this study. Data sharing is not applicable to this article.

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
