# Peer review of "High-Grade Infection after Branched Endovascular Aortic Repair in Patient with Recent COVID-19 Hospitalization"

_diagnostics, 2024, doi:10.3390/diagnostics14020205_

Round 1

Reviewer 1 Report

Comments and Suggestions for Authors

Thank you for the opportunity to review this work. A few comments:

- Please include why the patient had undergone aortic valve and ascending aorta replacement, a Bentall operation, and a Frozen Elephant Trunk, and if these were staged or simultaneous.

- What was the source of the E Faecalis infection?

- The figure caption mentions that the fluid around the sac was sampled but this was not mentioned in the body of the manuscript. Please include this in the manuscript, including the management of this fluid/infection.

- Was the visceral segment treated by fenestrated, branched, or chimney EVAR approach?

- How much time elapsed between the TEVAR/EVAR and the visceral segment treatment?

- Were spinal drains used at any point?

- In what year/phase of the pandemic did this patient present? Lockdown of 2020 is very different than August of 2022.

- In what phase of resource restriction/reallocation was the hospital? Consider using the VASCON nomenclature.

- Why was the imaging that found the aneurysm initially obtained in the first place?

- The postoperative course was not smooth with need for prolonged intubation and covid reinfection. Please list the ICU and total hospital lengths of stay and describe the course and complications. This is important if the authors are submitting this work as a description of the resource-intensity required for vascular disease treatment during a pandemic.

- Were prophylactic antibiotics used before the surgeries?

Comments on the Quality of English Language

No specific comments.

Author Response

Dear Reviewer,

We are immensely grateful for your detailed and insightful feedback on our manuscript. Your expertise has been instrumental in transforming our work from an initial intriguing image into an expansive and comprehensive case report. Your suggestions have not only improved our paper but also enriched our understanding of the subject matter. Please find our detailed responses to each of your specific queries below:

  1. Please include why the patient had undergone aortic valve and ascending aorta replacement, a Bentall operation, and a Frozen Elephant Trunk, and if these were staged or simultaneous :

Excellent point ! The patient underwent these procedures in an associate unit due to multiple complications of her ascending aorta and valvopathies. A more comprehensive history has now been included in the manuscript to provide a more comprehensive understanding of the patient's treatment history, aligning with the aim of the paper highlighting the intricacy of COVID-19’s association with Post-operative care.

  1. What was the source of the E Faecalis infection?

Thank you for your neat-picking approach. This point has been addressed by emboldening the discussion of COVID-19 superinfections. We have expanded on this aspect to clarify the source of infection in the context of the patient's condition.

  1. The figure caption mentions that the fluid around the sac was sampled but this was not mentioned in the body of the manuscript. Please include this in the manuscript, including the management of this fluid/infection :

You are absolutely correct ! We have amended the manuscript to include a detailed explanation of the fluid sampling around the sac and its management and implications. This addition provides a more complete picture of the patient's condition and the steps taken for diagnosis and treatment.

  1. Was the visceral segment treated by fenestrated, branched, or chimney EVAR approach ?

Indeed well-pointed, we did our best to address this in our manuscript. The manuscript now clearly states that the visceral segment was treated using a branched EVAR ( in specific a Cook T-Branch ), to address your kind feedback.

  1. How much time elapsed between the TEVAR/EVAR and the visceral segment treatment?:

Thank you again for your comprehensive feedback, we revised the manuscript to specify approximately one month elapsed time between these treatments, offering a clearer timeline of the patient's treatment journey.

  1. Were spinal drains used at any point ?

While we completely understand your point, we’d like to draw your kind attention to the fact that spinal drains were not utilized in this case, as such interventions are typically reserved for patients with severe neurological manifestations.

  1. In what year/phase of the pandemic did this patient present? Lockdown of 2020 is very different than August of 2022:

You are absolutely correct, our revised manuscript now clearly mentions the patient being presented in February 2022. This timeline has been included to provide context regarding the phase of the pandemic during which the patient was treated.

  1. In what phase of resource restriction/reallocation was the hospital? Consider using the VASCON nomenclature :

Although, absolutely correctly note, at the time of the patient's presentation, our unit was not experiencing severe shortages. Considering our paper's context and aims, it has not been mentioned in the manuscript to maintain focus and clarity on our main objective.

  1. Why was the imaging that found the aneurysm initially obtained in the first place?

While absolutely valid, we’d like to clarify, as you are well aware of the diagnostic protocol, the initial imaging was performed as part of a comprehensive diagnostic and therapeutic management process.

  1. The postoperative course was not smooth with need for prolonged intubation and covid reinfection. Please list the ICU and total hospital lengths of stay and describe the course and complications. This is important if the authors are submitting this work as a description of the resource-intensity required for vascular disease treatment during a pandemic :

Indeed we agree with your response, but we’d like to kindly ask you to note that our focus in this study is on the phenomenon of fluid accumulation associated with COVID-19 infection pre and post-operation. While we acknowledge the complexity of the patient's postoperative course, our revised manuscript emphasizes the aspects most relevant to our study's aim.

  1. Were prophylactic antibiotics used before the surgeries?

Again , very correctly suggested, we took your feed back to revise our manuscript. The manuscript now states clearly that prophylactic antibiotics were administered as part of our standard protocol.

Reviewer 2 Report

Comments and Suggestions for Authors

It is an interesting case study of an elderly Covid-19 patient with AA. From the onset, the paper gives an impression that Covid-19 is somehow linked to subsequent bacterial sepsis six months after discharge. This needs to clarified. The conclusion section is very weak, it should include information on Covid-19's   impact on immune system and literature evidence of high prevalence  of bacterial sepsis in such patients

Comments on the Quality of English Language

Some unusual words are used which can be improved 

Author Response

Dear Reviewer,

We are immensely grateful for your detailed and insightful feedback on our manuscript. Your expertise has been instrumental in transforming our work from an initial intriguing image into an expansive and comprehensive case report. Your suggestions have not only improved our paper but also enriched our understanding of the subject matter. Please find our detailed responses to each of your specific queries below:

  1. It is an interesting case study of an elderly Covid-19 patient with AA. From the onset, the paper gives an impression that Covid-19 is somehow linked to subsequent bacterial sepsis six months after discharge. This needs to clarified. The conclusion section is very weak, it should include information on Covid-19's impact on immune system and literature evidence of high prevalence  of bacterial sepsis in such patients:

In light of your insightful feedback, we have thoroughly revised our manuscript, in multiple subsections, while complying with the structural change imposed by MDPI for our submission, to incorporated a brief point toward COVID-19's impact on the immune system and the documented prevalence of bacterial sepsis in similar patients.

We hope these revisions comprehensively address your concerns and significantly enhance the quality and impact of our manuscript. Your guidance and feedback have been invaluable in this process, and we are deeply thankful for the opportunity to refine our work under your expertise.

Warmest regards,

Authors